# The Effect of Arabinoxylan and Wheat Bran Incorporation on Dough Rheology and Thermal Processing of Rotary-Moulded Biscuits

**DOI:** 10.3390/foods10102335

**Published:** 2021-09-30

**Authors:** María Teresa Molina, Lisa Lamothe, Deniz Z. Gunes, Sandra M. Vaz, Pedro Bouchon

**Affiliations:** 1Departament of Chemical and Bioprocess Engineering, Faculty of Engineering, Pontificia Universidad Católica de Chile, Avda. Vicuña Mackenna 4860, Macul, Santiago de Chile 7820436, Chile; mtmolin1@uc.cl; 2Nestlé Research Center, Institute of Material Science, Route du Jorat 57, 1000 Lausanne, Switzerland; Lisa.Lamothe@rdls.nestle.com (L.L.); zeyneldeniz.gunes@rdls.nestle.com (D.Z.G.); 3Nestlé Development Centre, Camino a Melipilla 15300, Maipú, Santiago de Chile 9260075, Chile; sandra.muntz.vaz@gmail.com

**Keywords:** rotary-moulded biscuit, wheat bran, arabinoxylan, elastic modulus, gelatinization, micro-CT

## Abstract

Wheat bran incorporation into biscuits may increase their nutritional value, however, it may affect dough rheology and baking performance, due to the effect of bran particles on dough structure and an increase in water absorption. This study analyzed the enrichment effect of wheat bran and arabinoxylans, the most important non-starch polysaccharides found in whole wheat flour, on dough rheology and thermal behaviour during processing of rotary-moulded biscuits. The objective was to understand the contribution of arabinoxylans during biscuit-making and their impact when incorporated as wheat bran. Refined flour was replaced at 25, 50, 75, or 100% by whole flour with different bran particle sizes (fine: 4% > 500 μm; coarse: 72% > 500 μm). The isolated effect of arabinoxylans was examined by preparing model flours, where refined flour was enriched with water-extractable and water-unextractable arabinoxylans. Wheat bran had the greatest impact on dough firmness and arabinoxylans had the greatest impact on the elastic response. The degree of starch gelatinization increased from 24 to 36% in biscuits enriched with arabinoxylans or whole flour and coarse bran. The microstructural analysis (SEM, micro-CT) suggested that fibre micropores may retain water inside their capillaries which can be released in a controlled manner during baking.

## 1. Introduction

Wheat bran constitutes about 16% of the weight of the wheat grain. It is the main by-product of wheat milling and is mostly used for animal feed [1]. On a weight basis, it contains about 46% dietary fibre, which mainly consists of arabinoxylans (70%), cellulose (24%) and β-glucans (6%) [2]. Accordingly, wheat bran is a convenient and inexpensive source of dietary fibre and specifically of insoluble fibre, which has beneficial effects on human health. Said effects include the acceleration of intestinal transit time and prevention/relief of constipation, where coarse insoluble fibres (e.g., wheat bran) mechanically irritate the gut mucosa stimulating water and mucous secretion (as a defense mechanism to protect from abrasion), and gel-forming soluble fibres resist feces dehydration, resulting in hydrated, bulky and easy-to-pass feces [3,4]. Additionally, viscous-soluble fibres reduce the postprandial glycemic response due to an increase in chyme viscosity, which results in a slower degradation of complex nutrients into absorbable components and slower absorption glucose and other nutrients at the brush border [3]. Furthermore, viscous-soluble fibres may also reduce the serum total and LDL-cholesterol due to an increased binding of bile acids in the small intestine resulting in their subsequent excretion [5].

Whole wheat flour has been used to enrich baked products with bran fibre, including bread [6,7], cracker biscuits [8,9], and sugary biscuits [10,11,12]. Studies show that the quality of the final product is detrimentally affected by bran incorporation because it modifies the rheological properties of the dough. Lapčíková et al. [6] and Le Bleis et al. [7] observed that bran addition and fine bran particles disrupted the continuous gluten network, affecting dough expansion and producing bread with a low specific volume and high density. Li et al. [9] showed that bran and especially arabinoxylans limited hydration of gluten proteins, which keeps them from forming appropriate networks for gas retention, precluding dough expansion of saltine crackers during baking and reducing their volume.

Sugary biscuit doughs are quite different from bread or saltine-cracker doughs. Sweet biscuits are formulated with higher amounts of sugar (10–33%, total dough wet basis) and fat (6–21%, total dough wet basis), and with less water (5–10%, total dough weight basis) than bread or saltine-cracker doughs [13]. During sweet-biscuit making, the mixing process is commonly divided into two stages to reduce the contact between flour and water as much as possible and thus limit the formation of a gluten network before baking [14]. Although the gluten network is limited in these products, bran enrichment may have a considerable effect on the rheological properties and baking performance of the biscuit dough, thus impacting final biscuits attributes. Milling of bran has been applied as a pre-processing step before its incorporation into whole grain products because coarse bran particles are highly visible and substantially affect their appearance [5]. Sozer et al. [10] examined the effect of bran particle size and bran supplementation from 0 to 30% total weight (based on flour replacement) during processing of rotary-moulded biscuits. They showed that the incorporation of fine bran particles (68 µm, mean diameter) increased the hardness of biscuits and the degree of gelatinized starch after baking, while coarser bran (450 µm, mean diameter) did not produce a significant effect. Likewise, Sudha et al. [12] analysed the effect of bran enrichment (particle size lower than 150 µm) from 0 to 40% total weight (based on flour replacement) on some physicochemical properties of rotary-moulded biscuits. They noticed that biscuits became harder to break as wheat bran content increased, but no further explanation was provided.

As mentioned above, arabinoxylans (AX) are the most important non-starch polysaccharides found in whole wheat flour and refined flour, and they have the highest water holding capacity among the flour polymers. Native starch, damaged starch and gluten proteins can hold 0.30–0.45, 1.0–10.0 and ~2.8 g of water per gram of dry matter, respectively, while arabinoxylans can hold 4–10 g of water per gram of dry matter [15]. Most of the scientific knowledge about the effect of arabinoxylans on biscuits has been taken from correlations between biscuit dimensions and the solvent retention capacity (SRC) test [16,17]. The SRC is a solvation test for flours that can be used to study the functional contribution of glutenins, damaged starch and arabinoxylans to the flour-swelling behaviour [15]. Ram et al. [17] analysed the relationship between the SRC values of fifty Indian wheat cultivars and the sugar-snap quality, and detected a strong negative correlation (r = −0.78) between the SRC test (for arabinoxylans: sucrose SRC) and the diameter of sugar-snap biscuits, suggesting that the water absorption of arabinoxylans detrimentally affected the biscuit quality. By contrast, Duyvejonck et al. [16] examined the contribution of flour constituents to the SRC profile for nineteen European wheat flours and did not observe any relationship between SRC values and total (TAX), water-extractable (WEAX) or water-unextractable (WUAX) arabinoxylans. Regarding the isolated effect of arabinoxylans in biscuits, Pareyt et al. [18] analysed the replacement of flour or sugar by arabinoxylan oligosaccharides (enzymatically derived from wheat bran arabinoxylan) on the quality attributes of sugar-snap biscuits. They found that flour replacement by arabinoxylan oligosaccharides produced unacceptable biscuits, whereas sucrose replacement by arabinoxylans oligosaccharides (up to 30 g per 100 g of sucrose) resulted in biscuits that were similar in diameter and height to control biscuits, suggesting their potential role as a sucrose replacer and dietary fibre contributor. To better understand these phenomena, it would be of great interest to examine the contribution of arabinoxylans compared to the direct addition of wheat bran in the rheological behaviour and the structure of sweet biscuits during baking.

The incorporation of wheat bran to biscuits would increase their nutritional value, however, its addition may affect dough rheology and baking performance, impacting final biscuits attributes due to the effect of bran particles on dough structure as well as to an increase in water absorption, where arabinoxylans may play a pivotal role. Accordingly, the aim of this study was to analyze the effect of water-unextractable and water-extractable wheat arabinoxylans addition and the effect of wheat bran flour enrichment on the dough rheology and starch gelatinization and on the resultant microstructure of rotary-moulded biscuits.

## 2. Materials and Methods

Soft wheat was supplied by Molinera San Cristobal (Santiago, Chile). Soluble wheat arabinoxylan (P-WAXYM, lot 40601) and insoluble wheat arabinoxylan (P-WAXYI, lot 120801b) were purchased from Megazyme Ltd. (Bray, Ireland). Palm oil (42.7% palmitic acid, 40.8% oleic acid, 9.5% linoleic acid, 4.7% stearic acid, 2.3% others) was kindly provided by Team Foods Spa (Santiago, Chile). Granulated sucrose was obtained from IANSA (Santiago, Chile), soy lecithin from Cargill (Santiago, Chile), ammonium bicarbonate from Pistor AG (Jura-Nord Vaudois, Switzerland), sodium bicarbonate from Andimex S.A. (Santiago, Chile), monocalcium phosphate from Blumos (Santiago, Chile), and salt from K + S Chile S.A. (Santiago, Chile).

Soft wheat was subjected to a milling process using a RT-1 pin mill (FP Spomax SA, Poland) to obtain refined flour (RF) and the wheat coarse bran fraction (72% of particles > 500 µm, with a Sauter diameter = 790 µm). The flour extraction rate was 75.7%. Part of the wheat bran was subjected to additional milling using a Mikro-Pulverizer MP1 (Micron Powder Systems, Summit, NJ, USA) to reduce its particle size and obtain the fine wheat bran fraction (4% of particles > 500 µm, with a Sauter diameter = 307 µm).

### 2.1. Preparation of Flour Blends

Whole flours with different bran particle sizes were reconstituted based on the flour extraction rate (75.7%). The coarse or fine bran fraction was used to obtain whole flour with coarse bran (WFL) and whole flour with fine bran (WFS), respectively (the chemical composition of the different flours is presented in Appendix A). To analyse the effect of whole flour enrichment on biscuit-making, different flour mixes were prepared by replacing 25, 50, 75 or 100% of the refined flour with WFL or WFS. The flour blends were dry blended using a N50 Hobart mixer with a flat beater at a low speed (60 rpm) to homogenize the mixture of refined and whole flours.

In order to understand the influence of AX during biscuit-making by isolating them from the other components of bran, a second system was examined which consisted of blends of refined flour enriched with water-unextractable and water-extractable wheat arabinoxylans. Four model flours were prepared by adding the amount of WUAX and WEAX necessary to achieve the same amount of arabinoxylans found in flour blends with either 50% or 100% WFS or WFL (determined using the phloroglucinol colorimetric method, as explained in Section 2.2). These model flours were referred as 50MFS, 100MFS, 50MFL, and 100MFL, respectively.

### 2.2. Proximate Composition Analysis

The chemical composition of the flour was measured using the following analyses for starch, moisture, proteins, fibre, and lipids [19,20] (See Appendix A).

The arabinoxylan content of flours was determined using the phloroglucinol colorimetric method, as described by Ramseyer et al. [21]. To determine the total arabinoxylan (TAX) content, 125 mg of flour was hydrated with 25 mL of distilled water in a 50 mL conical centrifuge tube. The sample was suspended by vortexing for 10 s. Then, 1 mL aliquot was immediately removed from the suspension and transferred into a reaction tube (30 g borosilicate glass, Anton Paar GmbH, Graz, Austria) using a 5-mL pipette tip. The total volume was brought to 2 mL by adding distilled water. To determine the water-extractable arabinoxylan (WEAX) content, 125 mg of flour was suspended with 25 mL of distilled water by vortexing. The suspension was placed in an orbital shaker (3D mini-shaker, Boeco, Hamburg, Germany) for 30 min at room temperature (23 °C). Afterwards, it was centrifuged at 2500× *g* for 10 min, and 1 mL supernatant aliquot was transferred into a reaction tube. The reaction agent was prepared by mixing 110 mL of glacial acetic acid, 2 mL of concentrated hydrochloric acid, 5 mL of 20% *w*/*v* phloroglucinol in absolute ethanol, and 1 mL of 1.75% *w*/*v* glucose. Then, 10 mL of the reaction agent was added to each reaction tube containing the samples for TAX and WEAX measurements. The tubes were placed in a boiling water bath for 25 min followed by rapid cooling in an ice bath up to 23 °C, and they were externally covered using aluminium foil. The absorbance of the sample was read at 552 and 510 nm, and these values were subtracted to remove the influence of hexose sugars. This was done immediately after cooling, as the absorbance may decrease after 10 min [22]. A standard curve was obtained using D-(+)-Xylose (Sigma-Aldrich, Santiago, Chile). A solution of 100 mg of xylose was prepared in 100 mL of distilled water (solution A). Aliquots of 0.02, 0.05, 0.1, 0.2, 0.3, and 0.4 mL were taken from solution A, and the volume was brought up to 2.0 mL with distilled water. Thus, standard solutions of 0.01, 0.025, 0.05, 0.1, 0.15, and 0.2 mg of xylose per ml of water were obtained. These standard solutions were subjected to the same procedure as flours (in triplicate). The arabinoxylan content was calculated using the xylose standard curve (y = 3.4895x, R^2^ = 0.992) and it was expressed as mg xylose equivalent. The water-unextractable arabinoxylan (WUAX) was obtained as the difference between TAX and WEAX.

### 2.3. Water Retention Capacity of Flours

Water retention capacity (WRC) was measured in triplicate using the procedure described by Hemdane et al. [23]. Flour (1.0 g) was suspended with 10 mL of distilled water in a 50-mL conical centrifuge tube by vortexing for 20 s. The sample was soaked for 60 min at room temperature (25 °C), followed by centrifugation at 4000× *g* for 10 min. The supernatant was carefully discarded, and the pellet was drained for 15 min. Finally, the pellet in the tube was weighed, and the WRC was quantified by subtracting the initial mass of the sample and tube. The results were expressed as g water retained per g dry sample.

### 2.4. Biscuit Dough Preparation

The standard dough was formulated according to Molina et al. [24] with some modifications, by mixing RF (65.2%, d.b.), sucrose (19.1%, d.b.), fat (13.1%, d.b.), monocalcium phosphate (0.9%, d.b.), sodium bicarbonate (0.7%, d.b.), ammonium bicarbonate (0.6%, d.b.), soy lecithin (0.3%, d.b.), salt (0.1%, d.b.), and water (17.7% w.b.). The fibre-enriched dough was formulated maintaining the same proportion of all ingredients, only replacing RF with each flour blend. Model-flour dough was prepared using the same ingredients proportion, considering the amount of AX to be added within the RF fraction. In this case, the moisture content of the dough was slightly increased to 20.2% (w.b.) or 22.7% (w.b.), in 50 MF and 100 MF samples, respectively, to obtain a mouldable dough.

The biscuit dough was prepared in two steps: the creaming phase and the dough phase. During the first step, sucrose, fat, water, soy lecithin, salt, sodium bicarbonate, and ammonium bicarbonate were mixed for 5 min in a N50 Hobart mixer using a flat beater at medium speed (124 rpm) as part of the creaming phase. Thereafter, the flour and monocalcium phosphate were added, and the mix was blended for 2 min at low speed (60 rpm) to obtain the biscuit dough. The dough was sheeted to a thickness of 3 mm, moulded in rectangular moulds (53 mm × 34 mm), and then baked at 170 °C for 13 min until a final moisture content of 2.3 ± 0.3% (w.b.) was reached.

### 2.5. Rheological Characterization

Dynamic oscillatory rheological measurements of biscuit doughs were conducted on a Physica MCR 501 rheometer (Anton Paar GmbH, Graz, Austria) following Ahmed et al. [25] with some modifications. The geometry of the measuring system was serrated parallel plates with 50 mm plate diameter. Before measurements were taken, the moulded dough was allowed to rest for 20 min for relaxation. Strain sweep tests (at 25 °C) were firstly conducted at a frequency of 1 Hz over 0.001–0.1% strain range to ensure that all other measurements were within the linear viscoelastic regime. Then, frequency sweep tests (at 25 °C) were performed in duplicate (2 batches per condition) from 0.01 to 10 Hz at a strain of 0.005% to determine the elastic modulus (G’), the loss modulus (G’’), and the loss tangent (tanδ = G’’/G’) as a function of frequency.

### 2.6. Texture Measurements

The firmness of the biscuit doughs was determined using a penetration test in a TA.HDplusC Texture Analyzer (5 kg loadcell, Stable Microsystems, Godalming, United Kingdom), following the procedure described by Colakoglu et al. [26]. The dough sample (~100 g) was weighed in a dough preparation set (A/DP) to remove the randomly distributed air inside the sample and to get a flat surface before the penetration test. The measurements were carried out using a P/6 cylindrical probe of 6 mm diameter. The probe penetrated the dough 20 mm at a test speed of 3 mm/s. Six replicates were taken of each batch, and this procedure was completed twice (2 batches per condition). The positive force value (N) at the maximum penetration depth was taken as a texture parameter of the dough firmness.

The hardness and fracturability of the biscuits were measured two days after they were made using a TA.XT Texture Analyzer (5 kg loadcell, Stable Microsystems, United Kingdom). Twenty biscuits per batch (3 batches per condition) were analysed using a three-point bending test (HDP/3PB probe) with a support span of 36 mm and test speed of 1 mm/s over a distance of 5 mm. The maximum breaking force (N) upon compression was used as a texture descriptor of hardness in biscuits, and the distance (mm) to the breaking point was used to represent fracturability [27].

### 2.7. Differential Scanning Calorimetry (DSC)

DSC analysis was performed in triplicate using a Mettler Toledo DSC823e (METTLER TOLEDO, USA) in order to quantify the degree of starch gelatinization (DG) in biscuits after baking following Molina et al. [28]. Rotary-moulded dough was dried at 43 °C for 20 h in an oven (T/UT 6200, ThermoFisher Scientific, Switzerland) so that it could be ground because crumbles of dough are difficult to hydrate with water and erroneous curves might be obtained. The ground sample (~16 mg, raw dough or baked biscuit) was placed in a 160 µL aluminium pan with distilled water ensuring a 1:4 ratio between the sample and the water. A pan containing distilled water (the same amount used for the sample preparation) was used as a reference. The pans were hermetically sealed and equilibrated at room temperature for 18 h prior to the analysis. The samples were heated from 25 to 90 °C at 5 °C/min, and the degree of starch gelatinization (DG, %) was quantified using the enthalpy values of raw ($∆H_{dough}$, J/g) and baked dough ($∆H_{biscuit}$, J/g), according to Equation (1):(1)$$DG\left(\%\right)=\left(\frac{∆H_{dough}-∆H_{biscuit}}{∆H_{dough}}\right)\cdot 100$$
where $\left(∆H_{dough}\right)$ corresponds to the enthalpy of all the starch granules embedded in the dough, which were forced to gelatinize in excess water, and $\left(∆H_{biscuit}\right)$ corresponds to the enthalpy of the starch granules that did not complete the gelatinization process during baking. Therefore, $\left(∆H_{dough}-∆H_{biscuit}\right)$ denotes the enthalpy of the gelatinized starch during baking. The enthalpy was expressed as J/g of dry starch in order to standardize the enthalpy values with respect to the amount of starch in the flour.

### 2.8. Microstructural Analysis

#### 2.8.1. Scanning Electron Microscopy (SEM)

Rotary-moulded biscuits were defatted prior to SEM analysis in order to reduce the fat interference during the visualization as suggested by Pareyt et al. [29]. The defatting procedure was carried out by soaking them in hexane (50 mL) for 30 min at room temperature. This process was repeated seven times and the solvent was replaced each time. The defatted biscuits were subsequently placed on a metallic stub equipped with double-side conductive carbon tape and tightened on stub with a silver conductive adhesive paste and a copper/nickel conductive tape. Before imaging, the biscuits were coated with a 10 nm gold layer using a SCD500 sputter coater (Leica Microsystems, Heerbrugg, Switzerland). Finally, the biscuits were examined using a Quanta F200 Scanning Electron Microscope (FEI Company, Eindhoven, the Netherlands) in low vacuum mode at 0.2 mbar by collecting secondary electrons with Everhart-Thornley Detector (ETD) and Back Scattered Electrons (BSE), at an accelerating potential of 5 kV with magnifications ranging from 200× to 1500×.

#### 2.8.2. X-ray Micro-Computed Tomography (X-ray Micro-CT)

The microstructure of rotary-moulded biscuits was characterized on a Skyscan 1272 X-ray computed microtomography system (version 1.1.17, Bruker Corp., Kontich, Belgium). The image acquisition was carried out with an X-ray source operated at a voltage of 40 kV and a current of 250 µA using an exposure time of 800 s per frame over an interval of 0–180° with a rotation step of 0.2° and two frames averaging on each rotation step. Three biscuits were scanned for each condition.

Around 962 projection images were obtained from the image acquisition, and they were reconstructed using an adapted cone-beam filtered back projection algorithm [30], and reconstruction software (NRecon v. 1.7.3, Bruker Corp., Kontich, Belgium). 2-D cross-sectional images were obtained with a resolution of 2016 × 1344 pixels and a voxel size of 5.5 µm × 5.5 µm × 5.5 µm. During the reconstruction step, critical parameters were set to obtain good quality reconstructed images: thermal correction (X/Y alignment with a reference scan), misalignment compensation (post-alignment), smoothing (1, using Gaussian Kernel = 2), ring artifacts reduction (=8), and beam-hardening correction (=40%).

### 2.9. Statistical Analysis

The experimental data were acquired in triplicate (3 batches per formulation) and analysed with R, version 3.6.1. For the statistical analysis, the bootstrap method introduced by Efron [31] was used as it makes no assumption about the underlying population distribution (i.e., a non-parametric approach). Accordingly, the analysis performed in this study did not assume that data come from a normal distribution and instead, fully accepted that the population distribution was unknown. The experimental data were used as an initial database to perform the analysis. Next, one thousand random replicates were extracted with replacement from the initial database to estimate the unknown population distribution of the statistical parameters of interest (mean in this case) and confidence intervals. Finally, 95% confidence intervals were used to test differences between the means of the different dough formulations.

## 3. Results and Discussion

### 3.1. Arabinoxylan Content and Water Retention Capacity of Flours

The WEAX and TAX fractions were determined in refined and whole flours (RF, WFL, and WFS). The TAX and WEAX contents were 2.20 ± 0.21% and 0.49 ± 0.05% in refined flour, 5.84 ± 0.44% and 0.51 ± 0.07% in whole flour with coarse bran, and 5.09 ± 0.25% and 0.55 ± 0.07% in whole flour with fine bran, respectively. The WUAX fraction, an important structural component in the cell wall of the endosperm and bran [32], constituted the greatest proportion of arabinoxylans. These results align with those of Kiszonas et al. [22] and Saulnier et al. [33], who reported that soft whole flour contains about 3.5–5.9% TAX and 0.3–0.6% WEAX, whereas refined flour contains about 2.2% TAX, and approximately 0.5% WEAX.

Table 1 shows the water retention capacity determined in refined flour (RF), whole flours (WFL and WFS), and model flours prepared with arabinoxylans (MFL and MFS). Overall, the water retention capacity increased as refined flour was enriched with whole flour or wheat arabinoxylans. A significant increase (*p*-value < 0.05) was observed in the water retained by flour blends enriched with whole flour, in particular at a replacement equal or over 50%. Bran particle size marginally affected this value; blends with fine bran particles retained ~2% more water than blends with coarse bran, but differences were not always significant. The water retention capacity in model flours prepared with arabinoxylans was significantly higher (*p*-value < 0.05) than in blends enriched with whole flour (with an equivalent arabinoxylans content). This shows that isolated fibre can retain more water than when it is part of the bran matrix. The WUAX/WEAX ratio in model flours increased from 3.5 (RF) to 7.5 and 6.3 in 50MFL and 50MFS, and to 10.9 and 8.7 in 100MFL and 100MFS, respectively. This suggests that both WUAX and WEAX, but especially WUAX, may influence the increase in flours’ water-retention capacity due to their strong tendency to absorb water, as discussed by Courtin et al. [34]. Also, dissimilarities in water retention capacity between whole flours and model flours might be explained by differences in the AX structure, particularly in the WUAX fraction. The three-dimensional conformation and the molecular heterogeneity of AX in wheat depends on the length of the xylan backbone, the arabinose-to-xylose ratio, the substitution and the distribution pattern, and the coupling of ferulic acids to other AX molecules of cell walls components [34]. WEAX are loosely bound to the surface of the cell wall, while WUAX are localized in the cell wall by covalent and non-covalent bonds with other cell wall components, such as proteins, lignin and cellulose [35], so that different extraction methods have been used to purify WUAX, such as barium hydroxide, alkaline hydrogen peroxide, and enzymatic methods [2,36]. Overall, commercial WUAX are treated to remove starch, β-glucans and proteins, and particular effort is devoted to maintain the ferulic acid crosslinks present in the native arabinoxylan [37]. This procedure could increase the water retention capacity of model flours, as AX fractions with a high ferulic acid content have shown to be capable of extensive crosslinking, yielding well-developed gel networks [35].

### 3.2. Firmness and Rheological Analyses of Biscuit Dough Enriched with Wheat Fibre

The viscoelastic properties of dough may influence the moulding stage and baking performance, impacting the final attributes of rotary-moulded biscuits. As such, firmness and rheological parameters were measured in fibre-enriched biscuit doughs. Figure 1 shows the firmness of doughs enriched with whole flour containing coarse or fine bran (Figure 1A) as well as those enriched with AX (Figure 1B). As reported in Figure 1A, the incorporation of whole flour significantly increased (*p*-value < 0.05) dough firmness compared to dough made with refined flour, which aligns with results reported by Nandeesh et al. [11] for rotary-moulded biscuits and by Filipčev et al. [38] for sugar-snap biscuits. Additionally, the bran particle size significantly affected (*p*-value < 0.05) this value. The firmness was ~8% higher in dough with fine bran particles compared to the dough with coarse bran at a replacement of 25%, and it rose ~17% at 50 and 75% substitution levels. The highest values of firmness were obtained in whole flour (i.e., 100% whole flour replacement), while dough containing finer bran had the greatest firmness. These results suggest that bran particles, especially smaller ones, produce a physical hindrance along the dough structure, as previously Sozer et al. [10] discussed regarding sugar-snap biscuit dough and Wang et al. [8] regarding cracker dough. For their part, arabinoxylans (WEAX and WUAX) had a lower impact on dough firmness compared to wheat bran, a behaviour that could be related to the amount of extra water that was added to achieve a mouldable dough. Besides, this parameter was more affected when doughs were prepared with model flours simulating 100% replacement (100MFL or 100MFS). Given that the effect of bran particle size is removed in these doughs, the firmness response is mainly influenced by the WUAX/WEAX ratio, where the WUAX fraction may contribute to increase the water holding capacity of the dough and the WEAX fraction mainly its viscosity [34].

A rheological analysis was performed to arrive at a better understanding about the effect of whole flour or arabinoxylan incorporation on the viscoelastic response of rotary-moulded dough. Figure 2 and Figure 3 shows the frequency sweep (G’/G’_control_) and loss tangent (G’’/G’) curves of doughs enriched with whole flours (Figure 2) or arabinoxylans (Figure 3). The storage modulus of each formulation was divided by the storage modulus of the dough prepared with refined flour (control) to better distinguish the effect of the enrichment on this variable. Overall, dough samples showed a predominant solid-like behaviour at the frequency range of 0.1 to 10 Hz, since the storage modulus (G’) was always higher than the loss modulus (G’’). This is consistent with the findings published by Li et al. [39] and Li et al. [9], who also observed an elastic-like behaviour (G’ > G’’) in doughs enriched with wheat bran. Furthermore, the elastic component of doughs gradually increased as they were supplemented with wheat fibre, but the values differed among samples containing different bran particle sizes or arabinoxylans. A rather similar elastic response was observed in doughs prepared with whole flours at 25% replacement (Figure 2A,B), with either coarse or fine bran particles. However, over 50% replacement, (G’/G’_control_) increased much more in doughs with fine bran particles. The different particle size of bran particles may explain these results, as finer bran particles can be better distributed along the dough structure compared to coarse bran particles, creating a more discontinuous structure (physical hindrance), which can increase the storage modulus [8,10]. A similar behaviour was observed in doughs enriched with arabinoxylans (Figure 3A), where the amount of AX increased the elastic response compared to the dough elaborated with refined flour. With respect to the loss tangent (G’’/G’), this value was significantly lower (*p*-value < 0.05) in doughs containing finer bran particles (Figure 2C), and it decreased much more when biscuit dough was enriched with arabinoxylans (Figure 3B), despite the higher amount of water that was added to obtain a mouldable dough. Several studies have suggested that bran or arabinoxylan addition causes considerable modifications to the gluten network in bread dough. Hemdane et al. [40] suggested that bran particles lead to the immobilization of water during dough resting, making the network progressively stiffer and, therefore, affecting the dough properties. Similarly, Li et al. [41] showed that water may migrate from the gluten to the arabinoxylan structure, detrimentally affecting the gluten network formation in whole wheat bread. In rotary-moulded dough, gluten development is importantly restricted due to the limited water availability, the high amount of sugar and fat, compared to bread dough, and the short mixing phase [28]. Accordingly, it is suggested that the rheological properties of biscuit dough might be affected by the physical hindrance of bran particles and the water retention capacity of arabinoxylans.

### 3.3. Starch Gelatinization during Baking and Microstructure of Fibre-Enriched Biscuits

In order to understand the influence of whole flour and AX on starch gelatinization, the degree of gelatinized starch (DG) of rotary-moulded biscuits was quantified. Results are presented in Figure 4. Overall, partial gelatinization (<38%) was observed for all biscuits, but the DG varied according to the enrichment with whole flour or arabinoxylans. In biscuits prepared with refined flour (control), only ~24.3% of native starch gelatinized during baking. This value rose significantly as coarse bran concentration increased, up to 36.3%, as shown in Figure 4A–WFL curve. However, no significant differences were observed in samples prepared with whole flour with fine bran particles compared to refined flour except from a minor significant increase (3%) at 75% replacement (Figure 4A–WFS curve). The increase in starch gelatinization related to bran inclusion may be due to the high water-retention capacity of bran, where water can slowly be released and make it available as cooking progresses [23,42]. In fact, Roozendaal et al. [42] concluded that bran can take up 200–500 times its own body mass in absorbed water (inside its micro-capillaries) and can release large amounts of water during baking. By contrast, Sabanis et al. [43] suggested that water retention of fibre may limit starch gelatinization. However, these authors did not consider the effect of the decrease in starch content due to wheat bran replacement when reporting the gelatinization enthalpy of dough enriched with wheat bran. This reduction is expected to diminish the gelatinization enthalpy as less starch is available in the system, affecting the reported results. In this study, the degree of gelatinization was quantified measuring the enthalpies of the dough and the baked biscuit prepared with the same formulation. As such, all values were standardized according to the amount of dry starch.

The difference in starch gelatinization due to bran particle size (Figure 4A) suggests that the microstructure of fibres may play an important role. To better examine this possibility, microstructural observations were performed using scanning electron microscopy (SEM) and X-ray micro-computed tomography (X-ray micro-CT), as shown in Figure 5. Figure 5a shows that the coarse bran fraction contains clear micropore arrangements, which are not observed when examining the fine bran fraction (Figure 5b), probably due to its destruction during extended milling, as previously exposed by Holopainen-Mantila et al. [44]. This micro-arrangement is also observed in rotary-moulded biscuits prepared with whole flour and coarse bran particles (Figure 5f, circled in yellow), and it is not detected in the control biscuit (Figure 5e) or in biscuits prepared with WFS (Figure 5g). X-ray micro-CT images (Figure 5j,k) show some pericarp fragments (highlighted in light yellow) with their characteristic micropores surrounding the air pores, in accordance with previous observations from Hemdane et al. [23], using SEM. These observations support the hypothesis that water could be held inside of these capillaries and then released in a controlled manner during baking. As explained by Jacobs et al. [45], bran capillaries may play a key role in enclosing water at micro- and nano-scales, mainly during the first 2.5 min of unconstrained absorption as long as the bran is not subjected to external forces. They also mentioned that the higher hydration properties of coarse wheat bran are attributable to weakly bound water located inside the micropores or between particles due to a less efficient stacking compared to fine bran particles. In the case of biscuits, the weakly bound portion of water may be removed from the coarse bran matrix during mixing, as the dough is constantly subjected to mechanical forces, and only the strongly bound water (inside the bran nanopores or through hydrogen bonding) may remain retained. However, after moulding, each moulded dough remains still on the baking tray for a few minutes (5 to 7 min, depending on the moulding rate) before the baking phase. It is suggested that this resting time may be enough for the coarse bran particles to rehydrate by reabsorbing the water in their micropores, while fine bran particles are not able to reabsorb water due to the lack of micropores. As heating takes place during baking, the water inside of the capillaries of coarse bran could be slowly released, allowing starch granules to partially undergo a thermal transition as cooking progresses.

Arabinoxylans’ direct inclusion also modified the degree of starch gelatinization (Figure 4B). Although DG was unaffected (*p*-value > 0.05) in biscuits prepared with 50MFS and 50MFL (mean DG of 23.6 or 23.7%), this value increased up to ~33.5 and ~36.3% when biscuits were formulated with 100MFS and 100MFL, respectively. These results show that the amount of arabinoxylans (WUAX and WEAX) affected the degree of starch gelatinization as well as the increase in the WUAX/WEAX ratio, as discussed in previous sections. Additionally, the DG of these biscuits (100MFL and 100MFS) was similar to the DG obtained in biscuits enriched with 100% of whole flour with coarse bran, despite the slightly amount of additional water in the dough. With regard to the microstructure, Figure 5c shows that water-extractable arabinoxylans (WEAX) appear as an agglomeration of nanoparticles, whereas Figure 5d displays water-unextractable arabinoxylans (WUAX) which contain micropore arrangements similar to those found in the coarse fraction of wheat bran. Furthermore, this microporous structure (circled in yellow) was also observed in the biscuit prepared with 100MFS or 100MFL, as shown in Figure 5h,i. As suggested for biscuits prepared with coarse bran, the porous structure of WUAX may explain the higher percentage of gelatinized starch in biscuits prepared with 100MFL or 100MFS as a result of the higher WUAX/WEAX ratio in these samples. Certainly, further research is needed to better understand this phenomenon.

SEM and X-ray micro-CT images showed that bran fractions were distributed along the biscuit matrix. This aspect may influence the final texture. Additionally, differences in dough rheology and thermal transition due to fibre enrichment were mainly observed at 50% whole flour replacement and above. To better understand this aspect on the texture of rotary-moulded biscuits, the hardness and the fracturability of biscuits prepared with 50 or 100% whole flour replacement were measured, as shown in Figure 6. The maximum breaking force in the bending test, which is reached upon breakage, is usually used as a texture descriptor of hardness in biscuits, and a higher peak force means a harder biscuit [46]. As shown in Figure 6A, hardness significantly increased when biscuits were enriched with whole flour, and biscuits with 100% replacement had the highest values. This aligns with the findings of Sudha et al. [12] and Jribi et al. [47], who showed that the incorporation of wheat bran increased the breaking strength of rotary-moulded and sugar snap biscuits, respectively. Bran particle size affected the hardness of biscuits at 50% replacement, but only 1 N difference was observed when comparing fine and coarse bran enrichment.

Enrichment with whole flour and bran particle size significantly affected (*p*-value < 0.05) the fracturability of biscuits, as shown in Figure 6B. This parameter is obtained from the force-distance curve and corresponds to the distance to the breaking point, where a lower value is related to a biscuit that is less compressible and more breakable [27]. Accordingly, biscuits with fine bran particles had lower fracturability than biscuits prepared with refined flour, as the distance to the breaking point increased from 0.25 mm (biscuit RF) to 0.30 mm (biscuit enriched with WFS). Biscuits enriched with whole flour and coarse bran had the lowest values of fracturability. Overall, these results indicate that a more compressible and less fracturable structure may be obtained when biscuits are enriched with bran, but the effect would be more pronounced when coarse bran particles are used.

## 4. Conclusions

This research focused on the enrichment effect of wheat AX and wheat bran with different particle sizes during processing of rotary-moulded biscuits. The results showed that refined flour enriched with wheat bran retained less water than when AXs were incorporated. Regarding the rheological properties, the firmness of the dough was affected by the wheat bran particle size. Smaller fractions produced a discontinuous and compact structure that increased the overall strength of dough compared to coarse bran particles. Additionally, the elastic response increased in dough prepared with bran or arabinoxylans, but it was even greater with the latter. This suggests that arabinoxylans contribute to the elasticity of biscuit dough while wheat bran contributes to its stiffness.

A partial starch gelatinization occurred in rotary-moulded biscuits, and a higher degree of gelatinized starch was obtained in biscuits enriched with AX or coarse bran particles. From a microstructural perspective, it was proposed that the micropores of coarse wheat bran or WUAX may retain water inside of their capillaries which could be released in a controlled manner during cooking, thus promoting this phenomenon.

Overall, the processing of rotary-moulded biscuit doughs was not affected by whole flour incorporation. The dough rheology was slightly modified with fine bran particles but did not alter the moulding stage, which is a critical part in this type of biscuits. Fine bran particles are a feasible option to elaborate rotary-moulded biscuits because their incorporation allows to control the degree of starch gelatinization and the fracturability of biscuits is practically unaltered compared to biscuits elaborated with refined flour. Accordingly, these biscuits may constitute a convenient and inexpensive source of dietary fibre.

## Figures and Tables

**Figure 1 foods-10-02335-f001:**
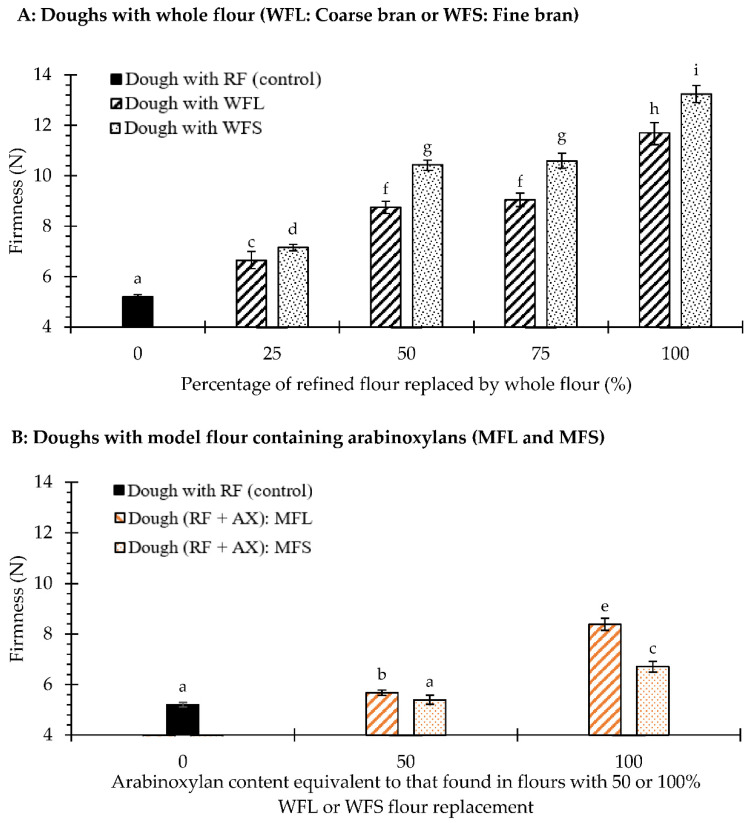
Firmness (N) of control dough, doughs prepared by replacing 25, 50, 75 or 100% of the refined flour with whole flour with coarse bran (WFL) or whole flour with fine bran (WFS) (**A**), and model flours enriched with arabinoxylan with an equivalent content to that found in flours with 50 or 100% whole flour replacement (MFL: coarse bran; MFS: fine bran) (**B**). Data are observed mean ± confidence intervals at 95%. Different superscript letters refer to significant difference (*p* < 0.05).

**Figure 2 foods-10-02335-f002:**
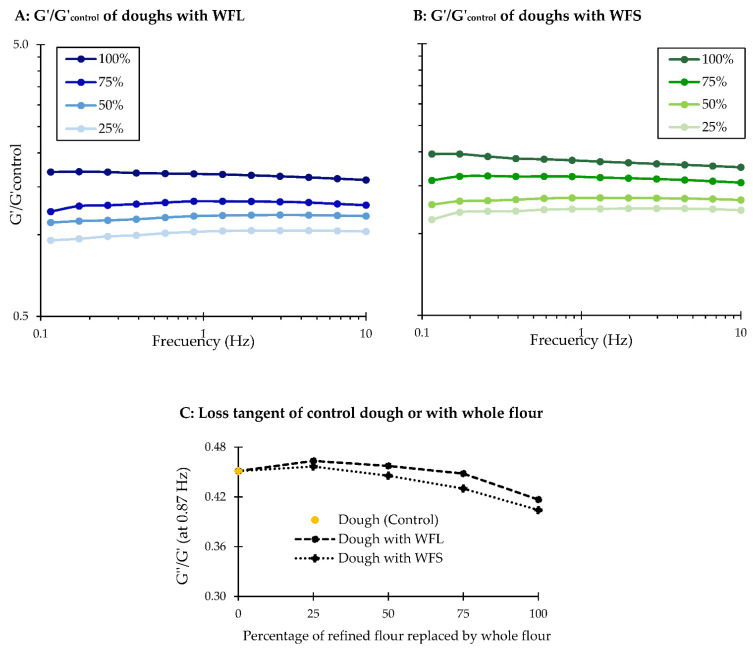
Frequency sweep curves represented by G’/G’control (where the control corresponds to the dough prepared with refined flour) as a function of frequency (Hz) of doughs prepared by replacing 25, 50, 75 or 100% of the refined flour with whole flour with coarse bran (WFL) (**A**) or whole flour with fine bran (WFS) (**B**). The loss tangent curves (G’’/G’) as a function of replacement at a frequency of 0.87 Hz are also shown (**C**).

**Figure 3 foods-10-02335-f003:**
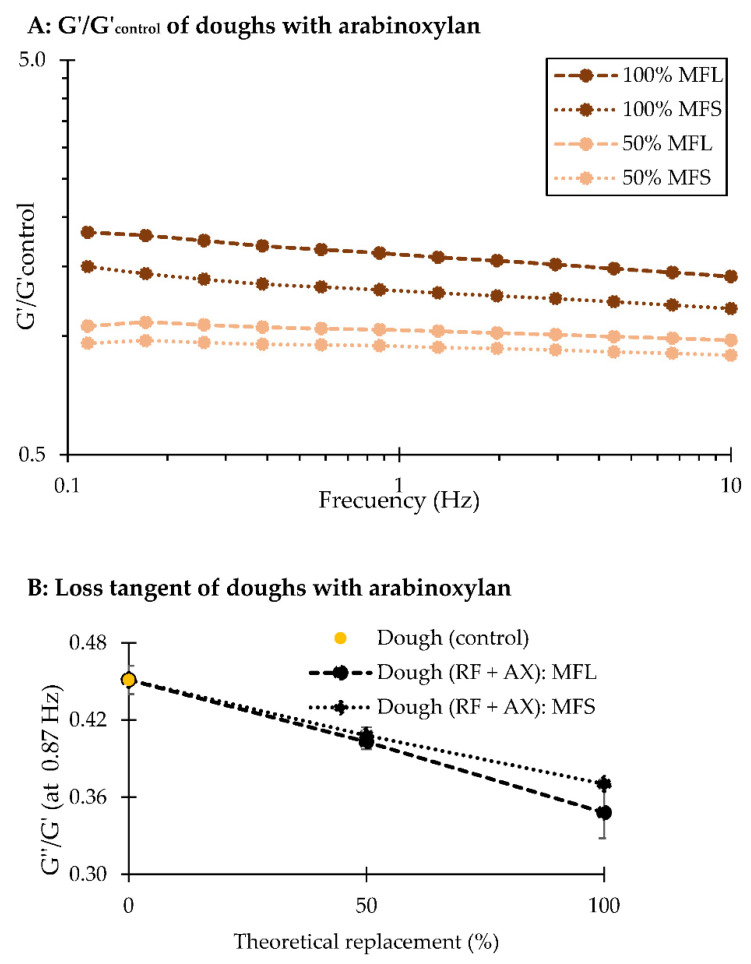
Frequency sweep curves represented by G’/G’control (where the control corresponds to the dough prepared with refined flour) as a function of frequency (Hz) of model flours enriched with arabinoxylans with an equivalent content to that found in flours with 50 or 100% whole flour replacement (MFL: coarse bran; MFS: fine bran) (**A**). The loss tangent curves (G’’/G’) as a function of replacement at a frequency of 0.87 Hz are also shown (**B**).

**Figure 4 foods-10-02335-f004:**
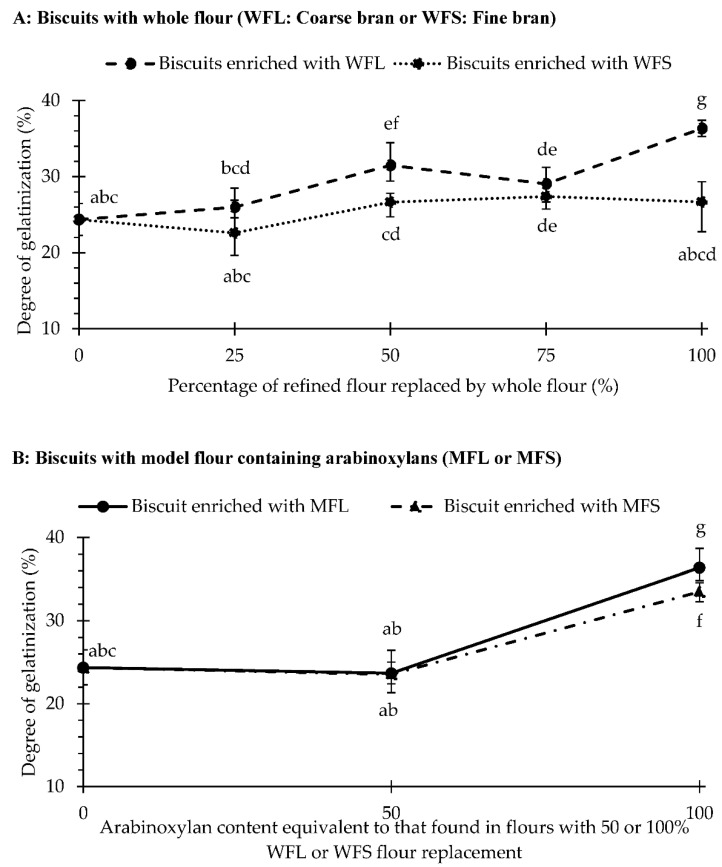
Degree of starch gelatinization (%) in rotary-moulded biscuits produced with refined flour, with doughs prepared replacing 25, 50, 75 or 100% of the refined flour with whole flour with coarse bran (WFL) or with whole flour with fine bran (WFS) (**A**), and with model flours enriched with arabinoxylan with an equivalent content to that found in flours with 50 or 100% whole flour replacement (MFL: coarse bran; MFS: fine bran) (**B**). Data are observed mean ± confidence intervals at 95%. Different script letters refer to significant difference (*p* < 0.05).

**Figure 5 foods-10-02335-f005:**
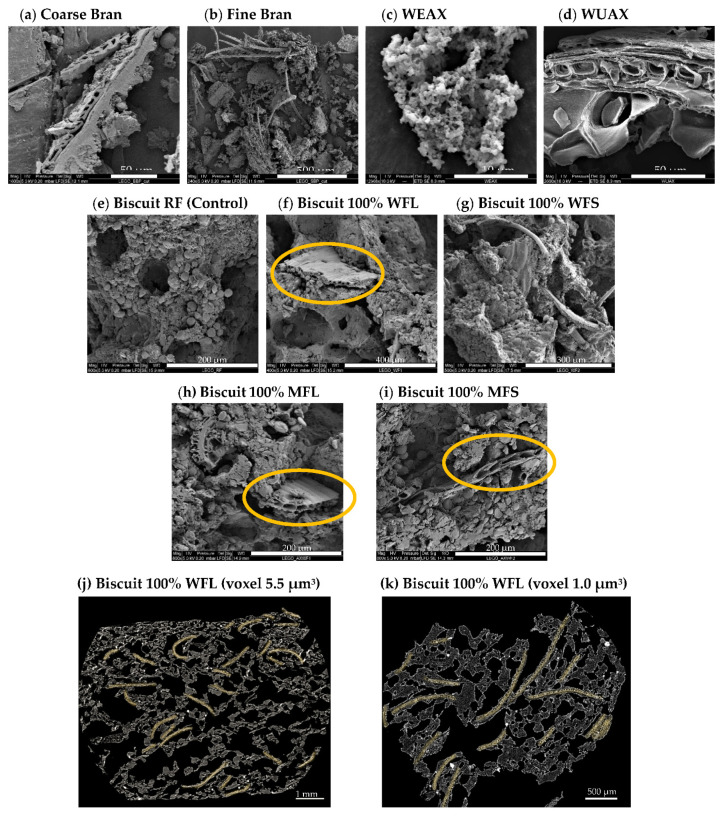
Scanning electron microphotographs (at magnifications between 240× and 12,968×) of the following fibres: coarse bran (**a**), fine bran (**b**), water-extractable arabinoxylan (**c**), and water-unextractable arabinoxylan (**d**), where the porous structure of the fibre is circled in yellow. Also, cross-sectional scanning electron microphotographs (at magnifications between 400× and 800×) of biscuits prepared with refined flour (RF) (**e**), whole flour with coarse bran (WFL) (**f**), whole flour with fine bran (WFS) (**g**), and model flours with an equivalent arabinoxylans content to that found in flours with 100% whole flour replacement (MFL or MFS) (**h**,**i**). Images (**j**,**k**) are cross-sectional images (z plane) obtained by X-ray µCT of a biscuit prepared with 100% whole flour with coarse bran (WFL), which is highlighted in light yellow.

**Figure 6 foods-10-02335-f006:**
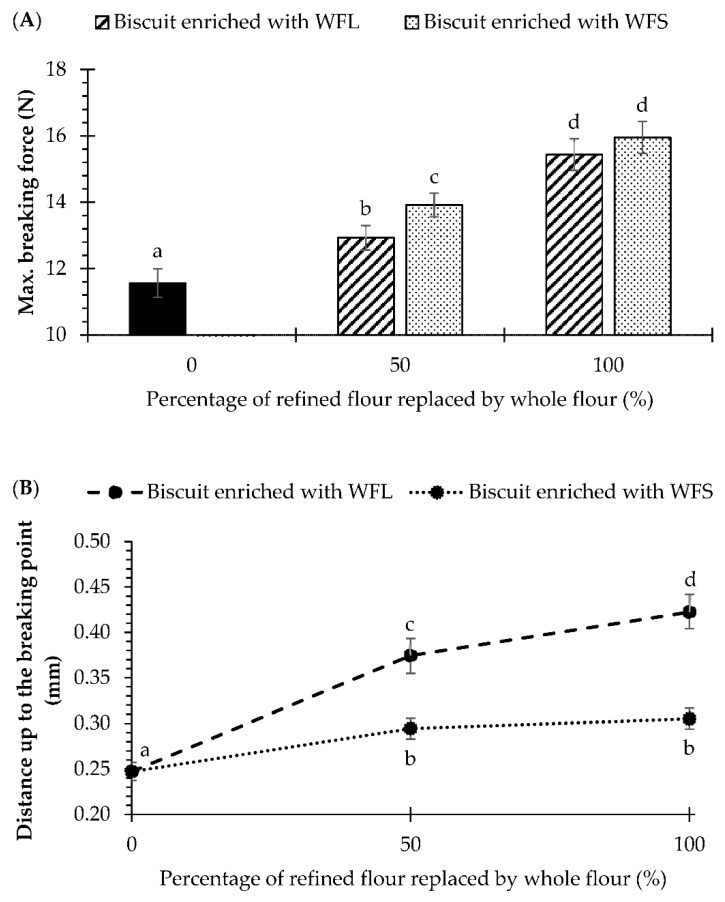
Maximum breaking force (**A**) and fracturability (**B**) of biscuits prepared with refined flour and enriched with whole flours containing coarse or fine bran (WFL or WFS) at 50 or 100% of replacement, respectively. Data are observed mean ± confidence intervals at 95%. Different scripts per parameter denote significant differences (*p* < 0.05).

**Table 1 foods-10-02335-t001:** Water retention capacity (expressed as g H_2_O/g dry sample × 100) measured in the refined flour (RF-Control), in the blends prepared by replacing 25, 50, 75 or 100% of the refined flour with whole flour with coarse bran (WFL) or whole flour with fine bran (WFS), and in model flours enriched with arabinoxylans with an equivalent content to that found in flours with 50 or 100% whole flour replacement (MFL: coarse bran; MFS: fine bran). Data are observed means ± confidence intervals (C.I.) at 95%. Different superscripts in “water retention capacity” column denote significant differences (*p* < 0.05).

Enrichment with Whole Flour	Flour Nomenclature	Water Retention Capacity (g H_2_O/g Dry Sample × 100)	C.I. at 95%
0%	RF-Control	64.1 ^a^	(60.6, 66.1)
25%	WFL	65.3 ^a^	(63.4, 66.5)
	WFS	66.4 ^a^	(66.4, 66.5)
50%	WFL	69.6 ^b^	(68.2, 70.7)
	WFS	71.3 ^c^	(70.9, 71.4)
75%	WFL	74.4 ^d^	(71.4, 76.0)
	WFS	75.4 ^d^	(73.5, 77.0)
100%	WFL	79.4 ^e^	(77.7, 82.0)
	WFS	81.9 ^f^	(80.8, 83.0)
**Enrichment with Arabinoxylans**	**Flour nomenclature**	**Water retention capacity** **(g H_2_O/g dry sample × 100)**	**C.I. at 95%**
50%	MFL	73.7 ^d^	(72.6, 74.3)
	MFS	72.2 ^bcd^	(70.4, 75.9)
100%	MFL	92.7 ^h^	(90.8, 95.4)
	MFS	86.5 ^g^	(86.2, 87.1)

## Data Availability

Data are contained within the article.

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
