# Peer review of "The Effect of Arabinoxylan and Wheat Bran Incorporation on Dough Rheology and Thermal Processing of Rotary-Moulded Biscuits"

_foods, 2021, doi:10.3390/foods10102335_

Round 1
Reviewer 1 Report
The manuscript titled “The effect of arabinoxylan and wheat bran incorporation on dough rheology and thermal processing of rotary-moulded biscuits” deals within the scope of the Foods Journal, by investigating an interesting topic of research. The quality of the presented research work is very good and represents valuable research results of the influence of arabinoxylan and wheat bran on the quality of biscuits.
Please find below some remarks to help the revision of the manuscript.
Line 102: Please, indicate the type of fat used.
Line 176: Authors should indicate the exact amounts of ammonium bicarbonate and sodium bicarbonate, not just the total amount of the leavening agent.
Line 189: Maybe it would be better to write: “The dough was sheeted to a thickness of 3 mm and moulded in rectangular molds (53 mm × 34 mm)…”
Lines 502-526: To gain a broader insight into the mechanisms of action of arabinoxylan it would be nice to present the results of the texture measurements for MFL and MFS biscuits, if any exist.
Author Response
Reviewer #1: General comments
The manuscript titled “The effect of arabinoxylan and wheat bran incorporation on dough rheology and thermal processing of rotary-moulded biscuits” deals within the scope of the Foods Journal, by investigating an interesting topic of research. The quality of the presented research work is very good and represents valuable research results of the influence of arabinoxylan and wheat bran on the quality of biscuits. Please find below some remarks to help the revision of the manuscript.
A: Thank you for your positive comments.
- Line 102: Please, indicate the type of fat used.
A: Thank you for the comment. The type of fat was included (lines 121-122), as suggested.
- Line 176: Authors should indicate the exact amounts of ammonium bicarbonate and sodium bicarbonate, not just the total amount of the leavening agent.
A: Thank you for the comment. The amount of each leavening agent was indicated (lines 199-200), as suggested.
- Line 189: Maybe it would be better to write: “The dough was sheeted to a thickness of 3 mm and moulded in rectangular molds (53 mm × 34 mm)…”
A: Thank you for the comment. The phrase “The dough was sheeted to a thickness of 3 mm and moulded in rectangular molds (53 mm × 34 mm)” was included (lines 212-213), as proposed.
- Lines 502-526: To gain a broader insight into the mechanisms of action of arabinoxylan it would be nice to present the results of the texture measurements for MFL and MFS biscuits, if any exist.
A: Thank you for this suggestion. We evaluated this possibility during the experimental design, however, it would have been a very expensive analysis because of the cost of pure AX and the number of replicates needed to obtain a reliable statistical analysis, so that we prioritize a microstructural characterization over the texture measurements on biscuit with MFL or MFS.

Reviewer 2 Report
Interesting work indeed!
- Abstract needs more work –
- add one or two sentences of on background
- What was your hypothesis
- What were the bran particle sizes used?
- Introduction:
- In the first paragraph, the health effect of cereal bran is beyond just acceleration of intestinal transit time and prevention/relief of constipation. Please give some metabolic examples and effect on gut microbiota as well with relevant references.
- I’d like to clearly see the hypothesis behind the aim of the study.
- Material and methods
- Section 2.2, first line: please cite the table S1 for the chemical composition of the flours, if that is the one you are referring to!
- Results:
- I could not understand the superscript in Table 1 at first. Please adda footnote as it is related with following figures in the manuscript!
- Conclusion:
- I’d like to see the prospective linkage between the results and the how it can contribute to human health outcome. How these results are impactful in that aspect?
Author Response
Reviewer #2: General comments
Interesting work indeed!
A: Thank you for your positive comments.
- Abstract needs more work –
- add one or two sentences of on background
- What was your hypothesis
- What were the bran particle sizes used?
A: Thank you for these suggestions. Background, hypothesis, and bran particle sizes were included, as proposed.
- Introduction:
- In the first paragraph, the health effect of cereal bran is beyond just acceleration of intestinal transit time and prevention/relief of constipation. Please give some metabolic examples and effect on gut microbiota as well with relevant references.
A: Thank you for the comment. Metabolic examples were included in lines 43-52 as well as relevant references, as suggested.
- I’d like to clearly see the hypothesis behind the aim of the study.
A: Thank you for the comment. The hypothesis was included (lines 110-113), as suggested.
- Material and methods: Section 2.2, first line: please cite the table S1 for the chemical composition of the flours, if that is the one you are referring to!
A: Thank you for the comment. The phrase “See Table S1 as supplementary material” was included (lines 155-156), as suggested.
- Results: I could not understand the superscript in Table 1 at first. Please add a footnote as it is related with following figures in the manuscript!
A: Thank you for the comment. The phrase Different superscripts in “water retention capacity” column was included, as suggested.
- Conclusion: I’d like to see the prospective linkage between the results and the how it can contribute to human health outcome. How these results are impactful in that aspect?
A: Thank you for the comment. A paragraph related with this was incorporated (lines 588-595), as suggested.
